# Are most published research findings false? Trends in statistical power, publication selection bias, and the false discovery rate in psychology (1975–2017)

Andreas Schneck [ORCID]*

Department of Sociology, Ludwig-Maximilians-University, Munich, Germany

* andreas.schneck@lmu.de

**Data Availability Statement:** All relevant data are within the manuscript and its Supporting Information files.

## Abstract

The validity of scientific findings may be challenged by the replicability crisis (or cases of fraud), which may result not only in a loss of trust within society but may also lead to wrong or even harmful policy or medical decisions. The question is: how reliable are scientific results that are reported as statistically significant, and how does this reliability develop over time? Based on 35,515 papers in psychology published between 1975 and 2017 containing 487,996 test values, this article empirically examines the statistical power, publication bias, and p-hacking, as well as the false discovery rate. Assuming constant true effects, the statistical power was found to be lower than the suggested 80% except for large underlying true effects ($d = 0.8$) and increased only slightly over time. Also, publication bias and p-hacking were found to be substantial. The share of false discoveries among all significant results was estimated at 17.7%, assuming a proportion $\theta = 50\%$ of all hypotheses being true and assuming that p-hacking is the only mechanism generating a higher proportion of just significant results compared to just nonsignificant results. As the analyses rely on multiple assumptions that cannot be tested, alternative scenarios were laid out, again resulting in the rather optimistic result that although research results may suffer from low statistical power and publication selection bias, most of the results reported as statistically significant may contain substantial results, rather than statistical artifacts.

## Introduction

The validity of scientific findings may be threatened by a poor research design or misconduct because the claimed effects may be only statistical artifacts, rather than existing effects [1]. Such statistical artifacts may have different facets, including non-transparent reporting of measures and samples [for a positive definition, see for experimental research 2, for observational research 3], but also the inadequate specification of sample size, which may be too small to identify an effect of interest (statistical power). Questionable research practices, such as erroneous rounding of results [4–6] and selective reporting of results that confirm the expectations

**Funding:** Funded by the German Federal Ministry of Education and Research, Quantitative Science Research (subproject Quantitative Analyses on Publication Bias, QUANT), grant agreement no. 01PU17009B. The funders had no role in study design, data collection and analysis, decision to publish, or preparation of the manuscript.

**Competing interests:** The authors have declared that no competing interests exist.

| Test | | Truth | | |
|---|---|---|---|---|
| | | Effect | No effect | |
| | Effect | True-positive (TP) | False-positive (FP) | **FDR** = FP / (TP + FP) |
| | No effect | False-negative (FN) | True-negative (TN) | |
| | | statistical power $\mathbf{pow}$ = TP / (TP + FN) | **FPR** = FP / (FP + TN) | |

**Fig 1. Truth table.** The truth table visualizes the different possible errors made by a statistical test: detecting a non-existing effect (false-positive (FP) rate (FPR)) or missing an existing one (1-pow). The FDR makes it possible to quantify the FP share out of all positive statistical effects (TP + FP).

[publication selection bias 6, 7], may pose a larger problem for validity because of the very low prevalence of wholly fabricated datasets [8].

This article shows the state and evolution of statistical power and publication selection bias in the form of publication bias or p-hacking in psychological articles published between 1975 and 2017. It also estimates the share of statistical artifacts on all statistically significant findings (hereinafter referred to as significant), called false discovery rate (FDR). The core research aim is to try to empirically answer the question posed by John Ioannidis: "why most published research findings are false" [2].

Statistical power is defined as the probability of finding a significant effect given that a true effect is present [9: 1]–displayed in the first column percentage in Fig 1. Usually, before data collection, researchers should determine their sample size to ensure the statistical power is as high as possible. Statistical power has hardly increased over time since the pioneering study by Cohen [10: 150f.] that examined the state of statistical power in leading psychological journals, and is still inadequate in the social sciences [11, 12] but also in other disciplines like medicine [13–15].

Low statistical power produces inefficiencies, as only a few existing effects can be detected. In this situation, the noise of the falsely detected tests (second column in Fig 1), defined by the significance threshold (equal to the FPR: e.g., set at 5%), drowns out the overall significant results. The FDR estimates the share of FP out of all significant tests detected by the test (first row percentage in Fig 1).

Jager and Leek [16] report a modest FDR of 14%, with no definite time trend, for the medical literature. This analysis, however, suffers from sample selectivity because only *p*-values reported in the abstract of the papers under study are used [for a critique, see 17]. An estimation based on the Open Science Center's replication project on the psychological literature predicted a much higher FDR of 38–58% [18] but is also restricted to a limited sample. Also, more theoretically derived estimates of the FDR yield much larger estimates, of at least 30% [19] or more [1].

Even larger FDRs are possible because p-hacking, a form of publication selection bias, may inflate the FP results [for simmulation studies, see 20: 1361, 21] and, as a result, the FDR. Publication selection bias is defined as reporting research results depending on their outcome, either in terms of direction (e.g., favoring positive outcomes) or significance [22: 135, 23, 24]. Publication selection bias is well-documented in various scientific disciplines [see, e.g., 7, 25] and may occur in two forms: publication bias and p-hacking. Publication bias is defined as publishing only studies with significant results, while studies with nonsignificant results remain in the file drawer and are therefore not published [26]. A bias closely related to publication bias is selective reporting bias, where significant results within a given dataset are reported, whereas nonsignificant results within the same dataset remain unpublished. Because publication bias and selective reporting bias cannot be distinguished in this study, the more

commonly used term "publication bias" is used. P-hacking, in contrast, searches for significant results within one dataset [20, 27], turning nonsignificant results into significant ones–in Fig 1, false-negatives (FN) to true-positives (TP) and true-negatives (TN) to FP. The practices of p-hacking comprise excluding outliers and switching outcomes, measures, or covariates to obtain a significant result [20]. While both forms of publication selection bias increase the FPR, they have different implications for the FDR. Publication bias does not alter the actual number of false-positive results, and therefore the FDR. P-hacking, in contrast, increases the FDR by transforming TN to FP (for theoretical considerations, see [1, 12]), as the resulting FP form a flood of statistical artifacts [28], outnumbering the TP effects.

## Data and methods

### Extraction of test statistics

The dataset at hand builds on all estimates reported in the text of empirical articles published in journals edited by the American Psychological Association (APA) between 1975 and 2017. The articles were identified and automatically exported from the bibliographical database *PsychArticles*. The automatic extraction routine covered full-texts in HTML and PDF form, using Optical Character Recognition (OCR) to recover the texts from the PDF files. It was possible to cover all reported test values in the articles' text that followed the strict publication manual published by the APA in 1974 [29] or later editions. The publication manual strongly recommends the in-text presentation of test values as a primary option if the test values are crucial to support the study's conclusions [30: 116f.]. A systematic extraction via regular expressions was possible because the APA publication manual has a strict reporting style that includes the kind of test statistic ($F$, $\chi^2$, $r$, $t$, $z$), its respective degree(s) of freedom, and its test value (e.g., $F(1,4) = 3.25$). An exception to this reporting style is the mandatory sample size (*N*) after the degrees of freedom for the $\chi^2$-test until the 6th revision of APA style [31–34]. The export routine allowed minor reporting inconsistencies in the articles (e.g., $X^2$ instead of $\chi^2$). Besides reporting test values directly in the articles' text, the publication manual also allows the use of tables to present results from more sophisticated research designs [31: 39]. However, 6th edition of APA style recommends using tables and figures for articles reporting more test values [>4 and >20 respectively 30: 116]. In contrast to in-text reporting, it was impossible to extract results reported in tables, due to the use of diverse reporting styles (e.g., standard errors or *p*-values being given below or beside the reported estimates). The in-text presentation of crucial results is, in contrast, consistent and varies only slightly over time (e.g., *N* reported along $\chi^2$).

Although the export routine covered relevant test values, it cannot distinguish between test values of substantial interest specified by hypotheses (the primary research outcome) and other reported statistical tests, especially diagnostic tests (see the section on robustness). Furthermore, it was impossible to identify whether (*t*, *r*, *z*) were used as one-sided or two-sided tests in articles. For the later computations, all of the aforementioned tests were assumed to be two-sided. The obtained test values were then used to compute the statistical power and to estimate publication selection bias.

### Computation of statistical power

To compute the statistical power, the true effect has to be specified, although by definition it is unknown. To this end, we used *a priori* plausible true effect sizes for small (*d* = 0.2), medium (*d* = 0.5) and large (*d* = 0.8) effect sizes [12, 35].

To calculate the statistical power based on Cohen's *d* metric, the standard error of the specific test has to be obtained. Using Cohen's *d* metric has an advantage over using other effect size metrics because the largest share of test values could be used for the analyses. The

conversion of the test values into standard errors in Cohen's $d$ metric was possible for $F$, $\chi^2$, both with two groups ($df_1 = 1$) and $r$, $t$ but not for $z$, because no information on the sample size was available. Multi-group comparisons for $F$ and $\chi^2$ ($df_1 > 1$) were therefore not included. For $F$, $\chi^2$, and $r$ the standard error was defined as: $\sigma_i = 2/\sqrt{N}$, whereas for $t$ an additional assumption on paired tests with equal group sizes and the correlation between observations ($r$) has to be made. As paired tests with a relatively high $r$ were the most optimistic (because smallest) estimate, $r = 0.5$ was assumed for all $t$. The standard error was then defined as: $\sigma_i = \sqrt{1/N}$. Besides Cohen's $d$ metric, the standard error of the Pearson correlation coefficient and the biserial correlation coefficients [36] were also calculated and transformed in Fisher's $z$ for the power estimation (see the section on robustness). Using other effect size metrics has the advantage that the specific assumptions concerning Cohen's $d$, which assumes a continuous outcome and a dichotomous assignment variable, could be cross-checked.

$$pow_i = \Phi\left(-th - \frac{d}{\sigma_i}\right) + \left(1 - \Phi\left(\text{th} - \frac{d}{\sigma_i}\right)\right)$$

Statistical power was defined as the probability of identifying a true effect (two-tailed cumulative distribution function of the standard normal distribution–first and second summand). Because the true underlying effect is, by definition, unknown, an *a priori* assumed true mean effect ($d$) is used. The test's precision is measured by the standard error of the effect size ($\sigma_i$) at the specified significance threshold (th, in this case, $z = 1.96$ for the 5% significance threshold).

## Estimation of publication selection bias

Publication selection bias was measured using the caliper test [CT, 37, 38, for similar approaches, see: 39]. The CT builds on the assumption that in a narrow interval (caliper) around the significance threshold, just significant results should be as likely as just nonsignificant results (50%–uniform probability in both calipers). The caliper width is defined relative to the significance threshold on the respective $z$-statistic (e.g., 5% caliper around the 5% significance threshold (1.96): ±0.05×1.96 around 1.96). The smaller the caliper is set, the more independent it is from the underlying test value distribution, but also the fewer values that can be utilized. A further advantage of wider calipers is the possibility of absorbing reporting inaccuracies (e.g., in rounding) as test values are reported only to two decimals. As simulations show [40], the 5% caliper around the 5% significance threshold offers a trade-off between both criteria. The 5% caliper contains the just significant $p$-values (so-called overcaliper, *oc*) in the range [0.039, 0.05] and the just nonsignificant $p$-values (so-called undercaliper, *uc*) from [0.05, 0.0626]. However the 10% [0.0311, 0.05], [0.05, 0.078] and 1% caliper [0.048, 0.05], [0.05, 0.052] were also estimated.

In contrast to previous studies, the CT in this article estimates the rate of publication selection bias, rather than serving as a diagnostic test. Although both forms of publication selection bias, publication bias and p-hacking, can be tested with the CT, they have different properties that must be considered when estimating their prevalence. Publication bias omits nonsignificant results while leaving the number of FPs unchanged. P-hacking, in contrast, converts nonsignificant results into significant results, increasing the number of FPs (cf. Fig 1) directly.

Although the CT can be used to estimate the rate of pure publication bias or p-hacking, it is not possible to identify if publication bias, p-hacking, or a combination of both strategies leads to an overrepresentation of just significant results (overcaliper). Consequently, it is only possible to estimate the resulting FDR for both strategies separately. In the case of no publication bias or p-hacking, the probability of just significant results (caliper ratio, CR) is expected to be 0.5 (even just significant and nonsignificant effects) because there should be no discontinuity

in the distribution of *p*-values around the rather arbitrary significance threshold. Assuming that the p-hacking rate is zero, the publication bias rate (*pbr*) is directly estimable from the ratio of the difference of the probability of just significant results, *CR*, and just nonsignificant, 1−*CR*, and *CR*, denoting the rate of dropped nonsignificant results. Simplifying the equation then yields:

$$pbr = \frac{CR - (1 - CR)}{CR} = 2 - \frac{1}{CR}$$

In contrast, estimating the p-hacking rate assuming publication bias is zero rests on more assumptions as nonsignificant studies are actively made significant by researchers (for a detailed description of the process of p-hacking, see Fig A in S1 Appendix). Therefore, the probability distribution of the respective hypotheses has to be recovered to estimate the needed p-hacking rate to fit the observed probability distribution of the just significant and nonsignificant values in the CT, given the estimated statistical power and the significance threshold. As this procedure involves multiple equations, a unit root solver was implemented to recover the p-hacking rate (*phr*, for further details on the estimation, see section 1.2., S1 Appendix).

Some assumptions have to be made for both estimators of publication bias and p-hacking. First, publication selection bias is implemented irrespective of the nonsignificant *p*-value achieved initially. This means that a strongly nonsignificant *p*-value is equally likely to be treated by publication selection bias and dropped with an equal probability in the case of publication bias as it is in the case of p-hacking (e.g., borderline significant results, $p = 0.051$ or far off $p = 0.9$). Second, p-hacking is always successful in finding a significant specification, as several rounds of drawing are performed until a statistically significant result is obtained. This second assumption on the actual form of p-hacking is not exhaustive because fine-tuning the same sample would be possible, e.g., by excluding outliers or changing measures or control variables. Such fine-tuning may produce a high share of just significant results even though the prevalence of underlying p-hacking is moderate. Third, although the CT does not require distributional assumptions when calculating the publication bias rate, the p-hacking rate depends on the assumed prevalence of true effects ($\theta$) and the estimated statistical power.

The truth table (Fig 1) works differently when considering publication bias. The cell frequencies of the TP and FP results are not affected by publication bias, as only nonsignificant results disappear in the file drawer (FN and TN results). The probability of finding statistically significant results that are true is not affected by publication bias. The probability of TP in the case of pure publication bias ($TP_{pb}$) is just the product of the prevalence of an underlying true effect ($\theta$) and the statistical power (*pow*). The same applies for $FP_{pb}$, which is defined as the product of no underlying true effect (1−$\theta$) and the set FPR ($\alpha$), in our case $\alpha = 0.05$. Both, $FN_{pb}$ and $TN_{pb}$, however, are reduced by the publication bias rate.

$$TP_{pb} = \theta \times pow$$

$$FP_{pb} = (1 - \theta) \times \alpha$$

$$FN_{pb} = \theta \times (1 - pow) \times (1 - pbr)$$

$$TN_{pb} = (1 - \theta) \times (1 - \alpha) \times (1 - pbr)$$

P-hacking works slightly different as nonsignificant results ($FN_{ph}$ and $TN_{ph}$) are replaced with significant results (that add to $TP_{ph}$ and $FP_{ph}$, rather than being dropped completely). $FN_{ph}$ and $TN_{ph}$ are therefore identical to the pure publication bias scenario, while the dropped

studies are added to $TP_{ph}$ and $FP_{ph}$, respectively. The unconditional probabilities are therefore:

$$TP_{ph} = \theta \times (pow + (1 - pow) \times phr)$$

$$FP_{ph} = (1 - \theta) \times (\alpha + (1 - \alpha) \times phr)$$

$$FN_{ph} = \theta \times (1 - pow) \times (1 - phr)$$

$$TN_{ph} = (1 - \theta) \times (1 - \alpha) \times (1 - phr)$$

From the estimated unconditional probabilities of TP, FP, FN, and TN for both scenarios of pure publication bias as well as pure p-hacking, the resulting FPR, statistical power, and FDR can be computed as laid out in Fig 1.

$$FPR = FP/(FP + TN)$$

$$pow = TP/(TP + FN)$$

$$FDR = TP/(FP + TP)$$

As it is impossible to assess the relative occurrence of p-hacking and publication bias from the published data, the two extreme scenarios of pure p-hacking without publication bias and pure publication bias without p-hacking were analyzed separately. The actual FDR may lie between these two extremes, assuming the simple mechanism of p-hacking and publication bias modeled is not too far from reality. Another assumption is the independence of tests, meaning that no (or not substantial) exact re-analyses are included in the data (studies using the same data and the same, or at least largely similar, analysis procedure), and there are no dependencies in the test statistics in one article (e.g., when reporting robustness checks). Furthermore, the mechanism of both publication bias and p-hacking is assumed to happen independently at the test level. Although publication bias may actually affect complete articles by leaving the entirety of results unpublished.

All three measures–statistical power, publication selection bias, and the false discovery rate–were aggregated yearly (see Tables F & G in S1 Appendix for robustness analyses of the statistical power on the test value level). In total, 43 observations, one for each year, are available. For the results on publication selection bias, the number of observations is reduced as each year had to contain at least estimates from 20 studies. Therefore, the dataset's early years before 1983 are missing, while 1981 is only present for the 5%- and 10%-CR. The results below were presented graphically via nonparametric LOESS regressions [40]. For parametric regression models, see S1 Appendix section 2.3.2).

## Results

### Sample construction

From the 53,860 empirical articles exported from PsycArticles, only 42,170 articles contained valid test values that could be transformed into 734,748 p-values (cp. Fig 2). There are two possible reasons for this: either no test value(s) were reported in an article (e.g., there were only descriptive analyses or graphical representations), or wrongly formatted test values were reported. In the second step, the standard error of each effect (test) was calculated in Cohen's $d$ metric to estimate the statistical power at specific *a priori* assumed small ($d = 0.2$), medium ($d = 0.5$), and large ($d = 0.8$) underlying effects. In this step, $z$-values and $\chi^2$ values with missing

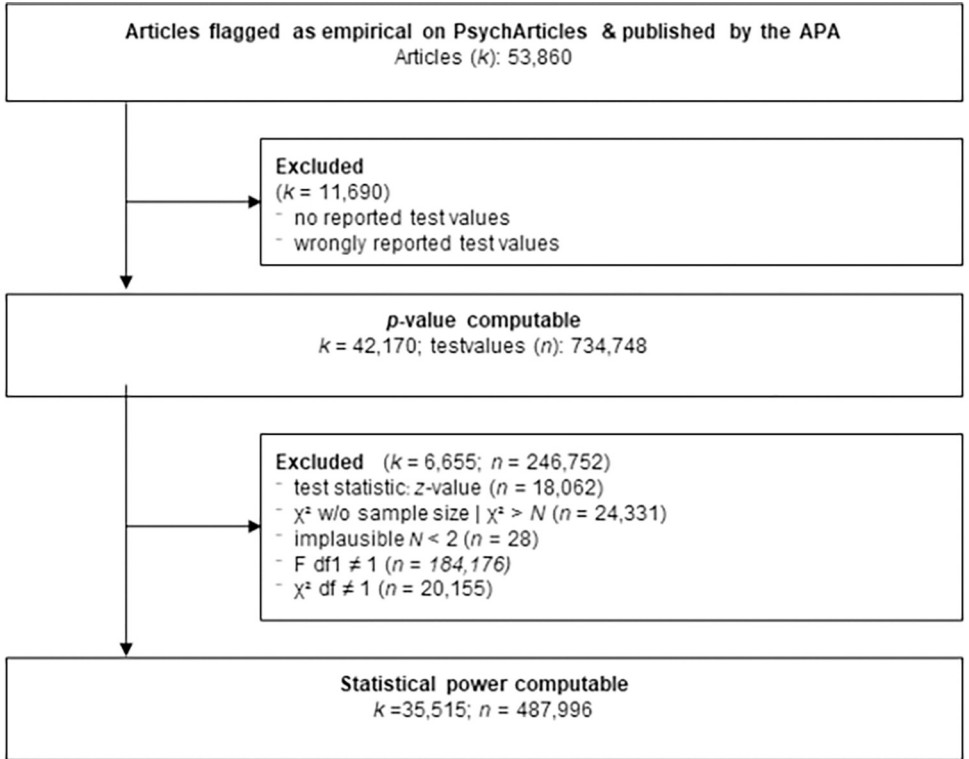

**Fig 2. Description of sampling and exclusion procedure.** Description of sampling process of articles as well as excluded test values.

or implausibly small ($N < 2$) sample sizes were dropped, as well as $\chi^2$ and F values with $df > 1$. After this step, 35,515 articles, containing 487,996 test values, remained in the analysis sample.

### Description of basic sample characteristics

The descriptive analyses of basic sample characteristics (cp. Table 1) can be structured on two levels, starting from the article level and finally moving to the level of reported test values.

**Table 1. Descriptive results.** Descriptive results for the different underlying meta-analyses, the included articles, and the test values.

| Type of observation | Number of observations | Mean | SD | Median | 1st quantile | 3rd quantile |
|---|---|---|---|---|---|---|
| **Articles** | | | | | | |
| Mean sample size for the tests | 35,515 | 504.0 | 51,082.0 | 61.5 | 31.0 | 133.0 |
| Nr. of reported test values | 35,515 | 13.7 | 14.4 | 9.0 | 4.0 | 19.0 |
| Article format | 35,515 | | | | | |
| . . . HTML | 33,780 | 95.1% | | | | |
| . . . PDF | 1,735 | 4.9% | | | | |
| **Test values** | | | | | | |
| Test statistic | 487,996 | | | | | |
| . . . $\chi^2$ df = 1 | 19,388 | 4.0% | | | | |
| . . . F df1 = 1 | 281,137 | 57.6% | | | | |
| . . . $r$ | 22,831 | 4.7% | | | | |
| . . . $t$ | 164,640 | 33.7% | | | | |
| Sample size for the test | 487,996 | 1,245.0 | 513,472.0 | 48.0 | 24.0 | 99.0 |

At the article level, the mean sample size over all reported tests in an article was extremely right-skewed (mean: 504.0, median: 61.5), mainly caused by studies with an extremely large number of observations. Compared with the mean sample size found in psychology, as documented in the literature [217.96, 211.03 and 195.78 in the years 1977, 1995, 2006, see 41: 338], the mean sample size in our study is more than twice as large (all years pooled: 504). Much smaller than the mean sample size in our study is the median sample size (all years pooled 61.5), which is also more comparable to the median sample sizes found in the psychological literature [48.40, 32.00, 40,00 in the years 1977, 1995, 2006, see 41: 338]. As the export routine intended to cover only relevant test values, an article's number of test values is of central interest. On average, 13.7 estimates (median: 9) were reported in the text of the article, with a maximum of 207 test values in one single paper. Because of these large numbers, robustness analyses were conducted for articles with up to five reported test values (see the section on robustness and section 2.3.1. in S1 Appendix). Of the included 35,515 articles, 4.9% could only be retrieved in PDF format, which could result in problems with misrecognized characters, and therefore test values. Articles in PDF format are especially problematic as they occur much more often in the years before 1985 and in the last two years, 2016 and 2017. However, in a robustness check comparing the years 2015 (10.2% PDFs) and 2016 (34.2% PDFs), there were no differences in the mean $p$-value, the CR used to estimate publication selection bias, and the type of test values between the neighboring years 2015 and 2016 that should be, except for the differences in article format, quite similar. However, the statistical power detecting a medium effect ($d = 0.5$) was statistically significantly lower for PDFs, by about 1 percentage point in 2015 and 2016.

On the level of test values, the $F$-test was the most-used test statistic, used in 57.6% of all tests. This descriptive result mirrors the large share of experimental studies using ANOVA analysis. $t$-tests were used in 33.7% of the reported test values. $\chi^2$-tests or $r$ were used only in 4.0% and 4.7% of the tests. The sample size for the test was, on average, 1,245 (median: 48), and thus larger than the mean on the article level, as one test value was based on more than 270 million observations (in this case, a manual check revealed that these were data on distinct human names from process-produced data).

73% of the test values were significant at the 5% significance level. When looking at the distribution of $p$-values in Fig 3 panel A, one can see that most of the test values were even significant at the 1% (49.8%) or even 0.01% significance threshold (23.6%). Fig 3 panel B, however, shows that around the 5% significance threshold using for descriptive purposes a caliper of 0.01 in terms of $p$-values (results close to the 5% caliper), there is a substantial jump, meaning that just significant values are much more common (in Fig 3, CR = 4.3/(4.3+2.4) = 64.2%) than expected in a uniform distribution of just significant and just nonsignificant results. This result was also highly statistically significant in a binomial test ($p < 0.00001$ for the 1%, 5% and 10% caliper), providing evidence for publication selection bias in the form of publication bias and/or p-hacking in the psychological literature. The presence of publication selection bias could also be seen when looking at different significance thresholds and stratifying the analysis by deciles of the underlying number of observations. Deciles with a smaller number of observations that were statistically significant at the 5% level were much more common than for lower thresholds (e.g., 1% and 0.1%), especially when compared with deciles with a larger number of observations (see Table B in S1 Appendix).

## Publication selection bias (publication bias and p-Hacking)

The publication bias rate, assuming no p-hacking, was 36.8% using the 5% caliper (cp. Fig 4A), meaning that more than one-third of the nonsignificant results may have disappeared in the file drawer before publication. In the alternative situation of p-hacking, assuming no publication bias

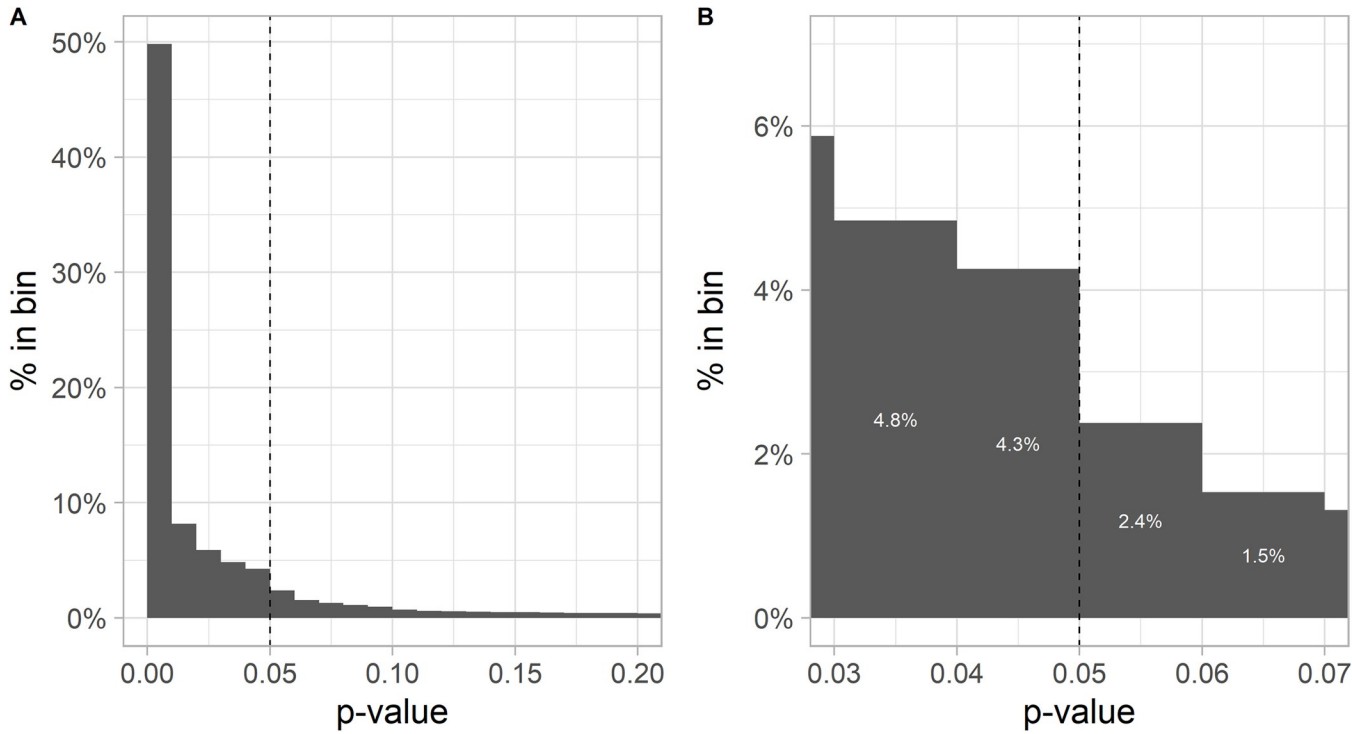

**Fig 3. Histogram of p-values from the test values.** Distribution of p-values in 0.01 binwidth. The y-axis denotes the probability of falling in the respective bin. This binwidth closely matches the 5% caliper around the 5% significance threshold. Panel A shows the distribution in the range between 0 and 0.2, higher values are dropped because of their rare occurrence. Panel B shows the bins around the 5% significance threshold. The results of the just significant (4%) and just nonsignificant caliper (2%) can then be used to calculate the publication bias and p-hacking rate.

(cp. Fig 4B), given the statistical power to detect medium effects ($d = 0.5$) and a 50% prevalence rate of a true effect (that results in half of the studies having a true effect of $d = 0.5$ and the other half $d = 0$), 8.2% of the results have to be turned significant to meet the observed caliper test results. The estimated publication selection bias was slightly larger based on the 10% caliper (42.1% for pure publication bias without p-hacking, 12.9% for pure p-hacking without publication bias) and considerably smaller in the 1% caliper (23.1% for publication bias, 3.3% for p-hacking, again in pure strategies). The narrowest (1%) caliper, however, was very volatile over the examined years, which may be a result of the smaller sample sizes in this narrowest caliper. Under pure p-hacking, without publication bias, the FPR would nearly triple to 13.9%, instead of the *a priori* set FPR, $\alpha = 5\%$. The FPR in the case of publication bias without p-hacking was 7.7%.

Both measures of publication selection bias (publication bias and p-hacking) varied only in the early years before 1985 (cp. Fig 4), resulting from the low number of estimates in those years. In recent years, there was only a small but unstable decline for the 5% and 10% caliper (for parametric regressions cp. Tables D & E in S1 Appendix). The 1% caliper even shows a slight increase that is mostly an artifact of the high volatility of the estimator. The pattern was also stable over the six subfields of psychology (social, cognitive, developmental, clinical, experimental, others) despite social psychology showing an increase and cognitive psychology showing a steep decrease in recent years (cp. Fig C in S1 Appendix).

## Statistical power

The statistical power over the examined years between 1975 and 2017 was, under the assumption of an underlying medium true effect ($d = 0.5$), 59.3% (Fig 5). For small underlying effects

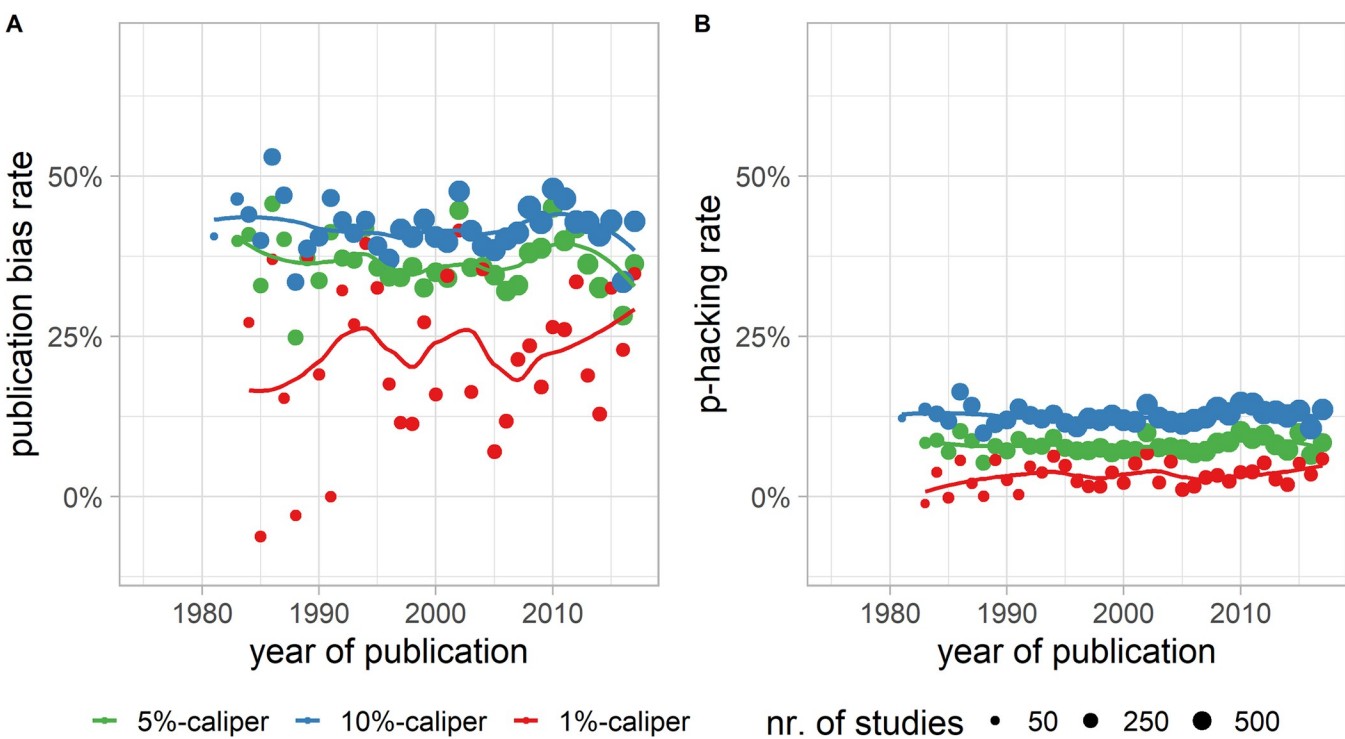

**Fig 4. Publication selection bias over time.** Estimated publication selection bias rate over time for different calipers (measures). Panel A shows the result assuming pure publication bias and no p-hacking, and panel B shows the results for pure p-hacking with no publication bias. It was technically impossible to model both p-hacking and publication bias at once. The dot size is determined by the number of studies (k) in the respective study year.

($d = 0.2$), the statistical power would be even lower, at 23.3%. Notably, only when choosing a large underlying true effect for all tests ($d = 0.8$) the statistical power crosses 80.4% the threshold of the recommended 80%.

The statistical power result shows that only slightly more than half of the existing medium ($d = 0.5$) effects were detectable with the research designs implemented in the original studies. However, after 2010 there was an upward trend in the statistical power to 68.3% for medium, 30.9% for small, and 85.8% for large underlying true effects in 2017 (for parametric regressions cp. Tables F—I in S1 Appendix). The large variation, especially before 1983, was caused mainly by the small number of included studies ($< 20$, denominated by a small point size in Fig 3). The upward trend of the statistical power was present in all psychological subfields. (cp. Fig D in S1 Appendix).

### False Discovery Rate (FDR)

In the following, the results of the FDR for a medium effect size ($d = 0.50$) and the 5% caliper bias estimate and a proportion of true hypotheses ($\theta = 50\%$) are presented (cp. Fig 6). Assuming pure p-hacking without publication bias, 16.2% of the significant effects would be false and wrongly detected, meaning those findings would be just statistical artifacts. Choosing a smaller proportion of true hypotheses $\theta$, the FDR increases ($\theta = 10\%$: 57.1% and $\theta = 20\%$: 38.4%). The inflated FDR ($FDR_{inf}$) was pronouncedly driven by p-hacking (Fig 6, red line), as the FDR without p-hacking (Fig 6, blue line) was, on average, 7.3% for $\theta = 50\%$, 41.5% for $\theta = 10\%$ and 24.0% for $\theta = 20\%$. This low FDR was the same for both no publication selection bias at all and pure publication bias. The time trend for the FDR estimates mirrored the trend of the

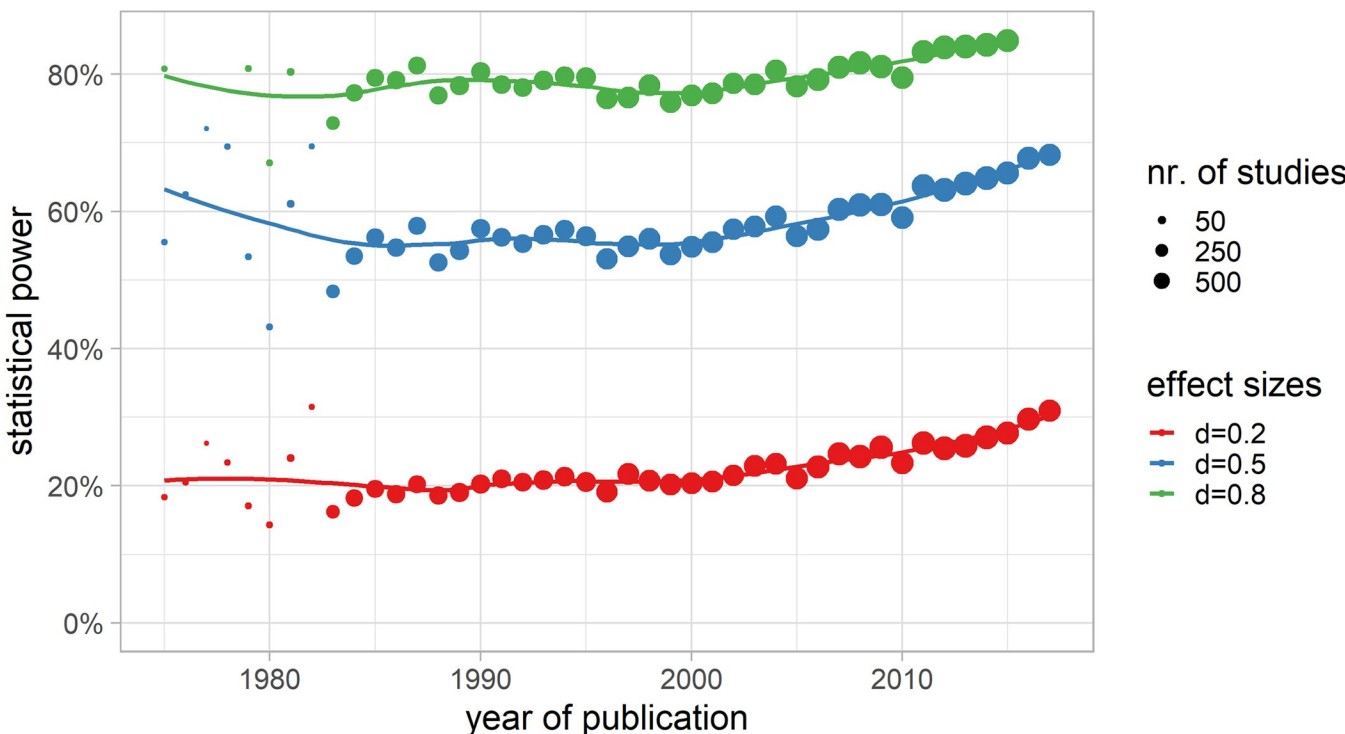

**Fig 5. Statistical power over time.** A priori statistical power for small (d = 0.2), medium (d = 0.4), and large (d = 0.8) underlying true effect sizes over time. The dot size is determined by the number of studies (k) in the respective study year.

statistical power and the p-hacking rate, showing a slight downward trend over time. Over the different subfields, the trend followed a similar pattern (cp. Fig E in S1 Appendix).

Up to this point, the FDR has only been reported based on the estimated statistical power and the 5% caliper p-hacking bias estimator. Given the high researcher's degrees of freedom in the analyses, a range of plausible alternative estimators is discussed below. In Fig 7, the proportion of true hypotheses ($\theta$) that is not estimable per definition makes up most of the variation in the FDR. The higher the share of true hypotheses $\theta$, the lower the FDR. 1- $\theta$ defines the maximum of the FDR because, under complete (100%) p-hacking bias, all estimates are artificially significant, irrespective of being true or not. The theoretical minimum of the FDR, given the 5% significance threshold, is used, and no p-hacking is present is $\frac{(1-\theta)0.05}{\theta pow+(1-\theta)0.05}$. The FDR assuming $\theta = 50\%$ is the most conservative setting but probably underestimates the actual FDR.

## Robustness

The robustness of the results was also tested for different effect sizes or sample specifications (see Table C in S1 Appendix). The sample was restricted to articles containing five or fewer test values in another robustness check. This more restricted sample may have contained test values that are relevant for the research question and have excluded irrelevant test values (e.g., manipulation checks or model fit statistics). This sample contained only 12,114 articles and 33,385 test values. All statistical power estimates were quite similar. In contrast, the share of just significant values from the caliper test for articles with fewer than five tests (5% CR: 0.646) was greater than for the main analysis (5% CR: 0.613). This may point to the fact that more primary research outcomes are covered in this sample. Using the standard error of the Pearson correlation coefficient (*r*) and the biserial correlation coefficientas as an effect metric also

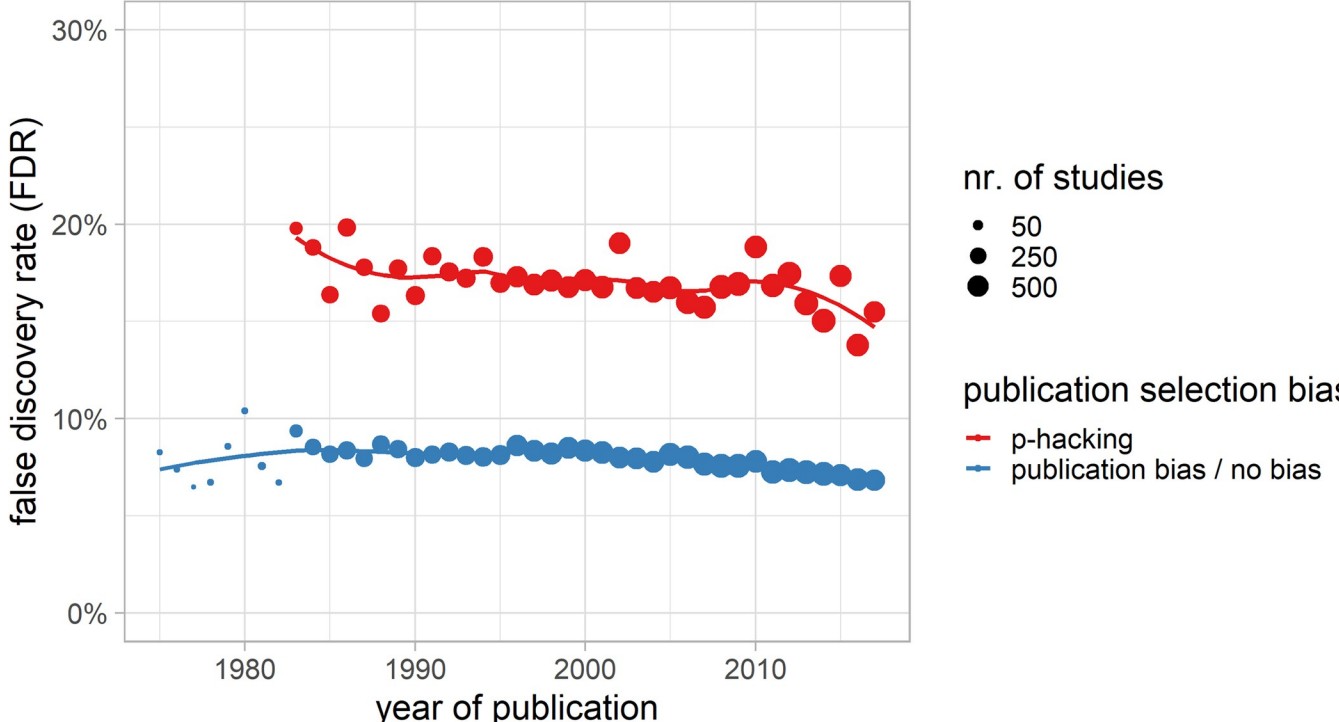

**Fig 6. FDR over time with and without publication selection bias.** Estimated false discovery rate (FDR) over time, either assuming pure p-hacking and no publication bias or pure publication bias and no p-hacking. For pure p-hacking or pure publication bias, the FDR is calculated from the caliper ratio (CR) based on a 5% caliper. The statistical power calculation was based on an a priori assumed medium effect size (d = 0.5). The proportion of tested hypotheses that are true (non-zero effects) was arbitrarily assumed to be θ = 50%. The FDR is mathematically identical if there is some pure publication bias or if there is no publication selection bias at all. For robustness for different measures, see Fig 7. The number of studies (k) in the respective study year determines the dot size.

showed no marked differences in the statistical power. When comparing the results between the different article formats in HTML and PDF, only negligible differences of a maximum of 0.8 percentage points occurred for all measures.

Although the assumptions needed for the test value calculation and computation of the statistical power could not be directly tested, it is possible to look at the results for the different test statistics. For publication selection bias, $\chi^2$-tests showed a higher CR (5%-CR: 0.638) than $F$-tests (5%-CR: 0.620), $t$ (5%-CR: 0.599) and $r$ (5%-CR: 0.586). These differences, however, seemed only to be minor. For the statistical power with a medium underlying true effect ($d = 0.5$), the by assumption paired $t$-tests (with between study $r = 0.5$) had by far the largest statistical power (81.0%), which may indicate that the assumption may have been overly optimistic. But $\chi^2$-tests also showed quite large statistical power (69.7%), followed by $r$ (52.0%) and the $F$-test (46.4%). Although a bias due to the assumptions of the effect size transformation cannot be ruled out, they should only lead to a minimal bias and more optimistic statistical power estimates.

## Central limitation

Besides the robustness of the results, which show no severe differences when using different subsamples, all presented results were inconsistent with the observed rate of significant results (73.0%). For example, the expected share of significant results with 59.3% statistical power finding a medium effect ($d = 0.5$), 8.2% pure p-hacking or 36.8% pure publication bias, and an assumed probability of 50% that the research hypothesis is true ($\theta$), would be 37.7% or 42.8%. Even in the most extreme case, assuming all research hypotheses are true ($\theta = 100\%$, expected

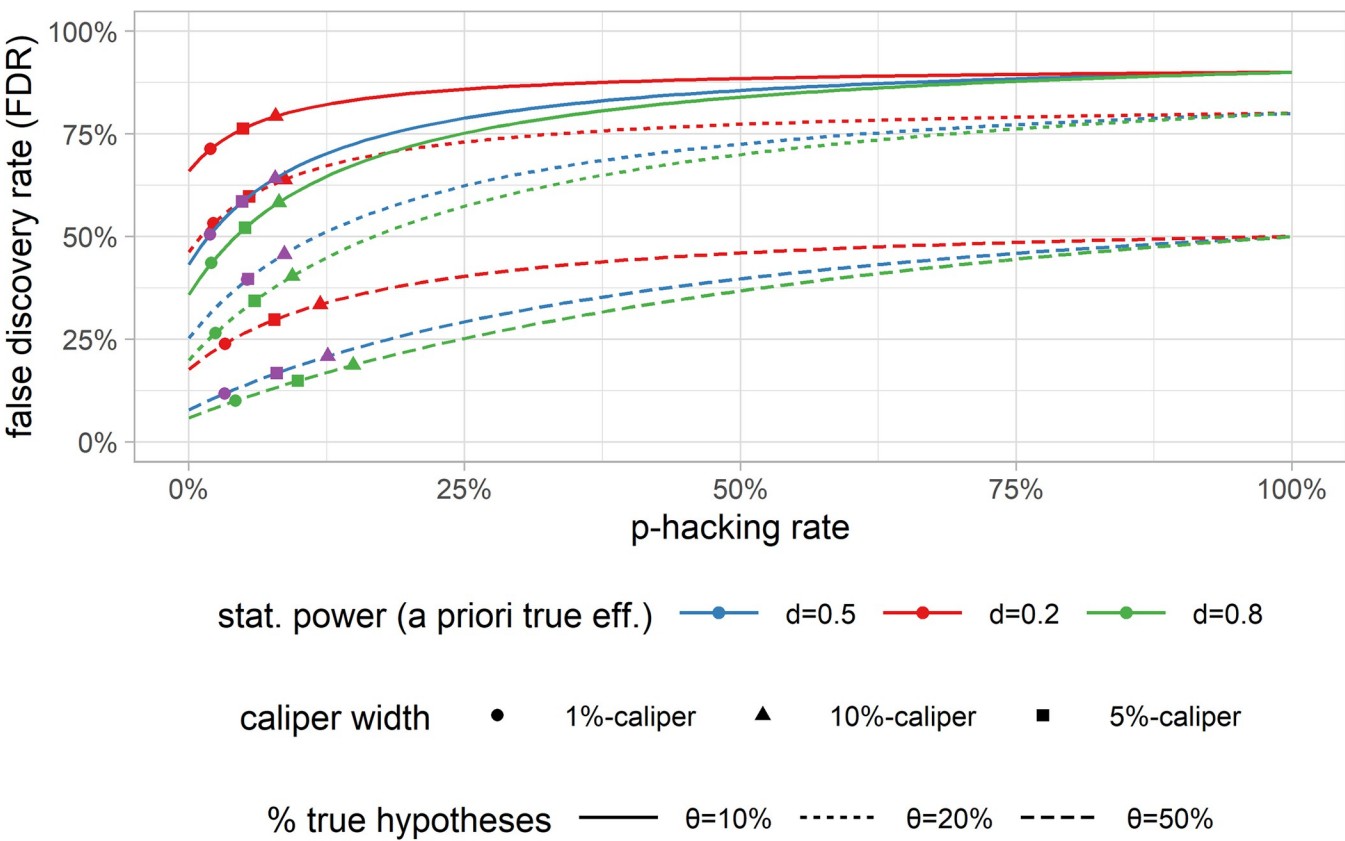

**Fig 7. FDR robustness plot over different proportions of true hypotheses, effect sizes, and p-hacking rates.** False discovery rate (FDR) over all possible p-hacking rates (0–100%), including empirical estimators of the p-hacking rate based on different calipers (1%, 5%, 10%) and different effect sizes (d = 0.2, d = 0.5 and d = 0.8) and for various theoretical proportions of tested hypotheses that are true (θ = 10, 20, 50%). The publication bias rate is assumed to be zero for calculating the p-hacking rate.

significant results 62.6% for pure p-hacking, 69.7% for pure publication bias), the observed share of significant results of 73.0% is still larger. Therefore, irrespective of the assumed share of true hypotheses, either the statistical power or the estimates on publication selection bias are biased.

In the case of an overestimated p-hacking rate, both statistical power and the prevalence of true effects must be relatively large (> 74%, cp. Table 2 Scenarios S1 & S2 –for a more flexible

**Table 2. Example of scenarios to fit the empirically observed share of statistically significant results P(p < 0.05) ~ 73.0%.** All hypothetical scenarios (S in columns) match the observed share of statistically significant results at the 5% significance threshold quite closely. The empirical estimates of the statistical power are highlighted in gray (d = 0.8 in S5 and d = 0.5 in S6 & S7).

| | Scenarios with share of statistically significant results ~ 73.0% | | | | | | |
|---|---|---|---|---|---|---|---|
| Parameter | S1 | S2 | S3 | S4 | S5 | S6 | S7 |
| Pure p-hacking rate | 0% | 0% | 10% | 22% | 30% | 60% | 70% |
| Pure publication bias rate | 0% | 0% | 13% | 32% | 40% | 72% | 96% |
| Power | 80% | 100% | 80% | 80% | 80.4% | 59.3% | 59.3% |
| θ | 91% | 73% | 87% | 80% | 75% | 50% | 10% |
| Resulting FDR | | | | | | | |
| FDR p-hacking | 0.6% | 1.9% | 2.6% | 7.1% | 11.5% | 42.6% | 88.0% |
| FDR publication bias | 0.6% | 1.9% | 0.9% | 1.5% | 2.0% | 5.9% | 43.2% |

graphical presentation, see Fig F in S1 Appendix). Because no publication selection bias is in place, the FDR is relatively low, at 0.6 or 1.7%. A slightly larger FDR from 0.9 to 7.1% was observed in scenarios with all adequately powered studies (Table 2 S3 & S4). When plugging in the empirically derived power estimates with, on average, 80.4% for large effects ($d = 0.8$) and 59.3% for medium effects ($d = 0.5$), considerable pure p-hacking (30–40%) or pure publication bias (40–72%) has to be in place to achieve the empirically observed significant results. In these scenarios, the FDR after pure p-hacking was 11.5% or 42.6%, and more than six times larger than in the scenario with publication bias or no publication selection bias. Finally, in an extreme scenario (S7), assuming a 10% proportion of true hypotheses and a 59.3% statistical power for medium underlying effects, the FDR was 88.0% with a high rate of pure p-hacking (70%), or 43.2% with pure publication bias (96%). Although these seven scenarios show a considerable range of possible FDR scenarios that produce the observed share of significant results, the author considers scenarios S5 and S6 the most plausible as psychological research is known to suffer from a quite low statistical power [12], and theoretical considerations regarding primary research indicate that the share of true hypotheses ($\theta$) should be rather low [1].

## Discussion and conclusion

This article has presented the state and evolution of statistical power and publication selection bias, in the form of publication bias and p-hacking, and the FDR over time in a sample of 35,515 papers in psychology published between 1975 and 2017. Despite a positive development in recent years, the statistical power to detect medium effects ($d = 0.5$), and therefore sample size, is still below the recommended 80%. Furthermore, in the data, patterns of publication selection bias (in the form of publication bias, p-hacking, or both) were detected but showed no substantial decline over time. The resulting $FDR_{inf}$ illustrates the consequences of low statistical power and p-hacking, as a considerable share of all significant results may be just statistical artifacts, and therefore false.

Two strengths of the analysis can be pointed out: first, the article draws from a full sample of empirical articles in journals published by the APA and the therein reported statistical test values, therefore it is equipped with high statistical precision due to the inclusion of many studies and test values. Second, the full sample provides good external validity, which makes it possible to generalize the results at least over the field of psychology.

Although the article gives a plausible range of possible measures of statistical power and publication selection bias, it also suffers, in addition to the central limitation pointed out earlier (i.e. the missing fit between the observed and expected share of statistically significant results), from five limitations. First, the parameter with the largest leverage on the FDR, the share of true hypotheses, is not estimable and remains open for speculation. Second, all APA style-compliant test values were exported irrespective of their relevance. According to the APA publication manual, although those results should be of central importance, they are not necessarily the primary research outcomes. A robustness check, comparing manual to automatic data extraction, shows that primary research outcomes were only poorly covered (for details, see section 2.1. in S1 Appendix). Third, estimating the statistical power assumes a constant underlying effect size across psychology, with no dependency on sample size and underlying true effect size. The unobserved heterogeneity may lead to the problem that hypotheses (e.g., with a large underlying true effect) were deliberately conducted with small, adequately powered samples but may have been classified as underpowered. Fourth, although publication bias and p-hacking were estimated separately, it is not possible to uncover their relative prevalence from the data. Fifth, the assumptions on the underlying process of publication bias and p-hacking have to be valid to give unbiased estimates of the FDR; this may be especially

problematic for p-hacking because a large number of assumptions have to be made (for details, see section 1.2. in S1 Appendix).

Keeping these five limitations–and especially the central limitation–in mind, a wide range of possible FDRs may be compatible with the observed share of significant results. In the most extreme cases, the FDR can range, as described in Table 2, from less than 1% to nearly 90%. However, besides the unsolvable modeling limitations of the central parameters of interest, publication selection bias, statistical power and the FDR, the article aimed to investigate the up to now completely theoretical considerations on the aforementioned parameters at least to some degree empirically.

A plausible range of the FDR therefore lies between the FDR with no bias (publication bias) and its version inflated by p-hacking. Although the article gives a plausible range of statistical power and publication selection bias estimators, the exact results should be interpreted with caution.

When comparing the estimated statistical power with previous studies in psychology, it was found to be slightly larger [11, 12], which could be caused by the more optimistic estimation strategy. Although publication selection bias has already been investigated in the literature, to my knowledge, no estimates of publication selection bias rates exist. The present study has also disentangled the impact of the estimated statistical power and publication selection bias on the FDR. The only study calculating a discipline-wide FDR [16] was replicated for conditions without the influence of p-hacking but provided a slightly larger FDR under p-hacking.

To sum up, findings reported in the psychological literature tend to be underpowered and shows signs of p-hacking or publication bias, which may lead to a high number of false discoveries; however, the estimations rely on several assumptions and unknown parameters that may influence the estimates. Nonetheless, three interventions can be laid out that may be of help to decrease the FDR and increase the credibility of scientific findings: first, conducting mandatory power analyses before a study; second, pre-registering the research design, along with a complete model specification; and third, developing clear-cut reporting guidelines on how to present statistical tests. While the first intervention focuses on maximizing statistical power, this will only be effective if real pre-study power analyses are conducted, instead of *post-hoc* power analyses that justify the chosen setting. These pre-study power analyses must justify, in particular, their assumed underlying true effect. Otherwise, pre-study power analyses may only justify a sample size already set before the power analysis. Besides promoting pre-study power analyses, pre-registration would make it harder to engage in publication selection bias [42]. To allow researchers to fix errors or improve their research design, deviations from the pre-registration should be possible, but should be laid out in the article. In contrast to mandatory power analyses and pre-registration, which would help increase statistical power or decrease publication selection bias, reporting guidelines can help in monitoring those measures. Building on this monitoring, interventions such as data-sharing policies of journals can be evaluated. In the case at hand, the results at hand were only possible because existing reporting guidelines for psychology allow for structured analyses. Other disciplines, like economics, medicine, or sociology, have no such guidelines. Therefore, there is no reason to believe that the diagnosed problems are limited to psychology–and especially to APA journals–alone. Beyond such descriptive monitoring, future research could focus on the mechanisms behind potential risk factors for a low statistical power or publication selection bias, such as author composition or third-party funding.

## Supporting information

**S1 Appendix. Additional descriptions of methods and analyses.**
(PDF)

**S1 Data. Dataset and scripts for replication.**
(ZIP)

## Acknowledgments

I gratefully acknowledge Katrin Auspurg, Stephan Bruns, Thomas Hinz, and Peter Pütz, for their comments on earlier versions of the article. I also want to thank the participants of the Metrics International Forum and the Seminar Analytical Sociology: Theory and Empirical Applications for their helpful remarks. Finally, I want to thank the two reviewers, Esther Maassen and André Gillibert, for their very detailed and valuable reviews. André Gillibert substantially improved the paper with his suggested estimation of publication selection bias.

## Author Contributions

**Conceptualization:** Andreas Schneck.

**Data curation:** Andreas Schneck.

**Formal analysis:** Andreas Schneck.

**Funding acquisition:** Andreas Schneck.

**Investigation:** Andreas Schneck.

**Methodology:** Andreas Schneck.

**Project administration:** Andreas Schneck.

**Resources:** Andreas Schneck.

**Software:** Andreas Schneck.

**Validation:** Andreas Schneck.

**Visualization:** Andreas Schneck.

**Writing – original draft:** Andreas Schneck.

**Writing – review & editing:** Andreas Schneck.

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
