## [Decision Letter · Decision Letter 0]

23 Mar 2021

PONE-D-21-02988

Are most published research findings false? Trends in statistical power, publication bias and p-hacking as well as the false discovery rate in psychology (1975–2017)

PLOS ONE

Dear Dr. Schneck,

Thank you for submitting your manuscript to PLOS ONE. After careful consideration, we feel that it has merit but does not fully meet PLOS ONE’s publication criteria as it currently stands. Therefore, we invite you to submit a revised version of the manuscript that addresses the points raised during the review process.

**First of all I would like to thank the two reviewers of this manuscript. **They did a fast a very comprehensive peer review. I found it all the more informative.

As far as I can tell, **there are major concerns with the methods **used in this manuscript, as suggested by reviewer 2. I cannot guarantee acceptance at this step but I will be happy to read a new edited version of the manuscript with all the appropriate responses. I expect sound responses before making a decision. Please fix any issue that can be fixed and please make sure you explain very carefully all your limitations in the paper (and also in the abstract (to avoid any spin)). Please also consider fixing the potential errors with the R code. Again, there is no guarantee that the edited version of the manuscript will be accepted after resubmission as there are many concerns but I would be happy to read an improved version of the paper.   

We look forward to receiving your revised manuscript.

Kind regards,

Florian Naudet, M.D., M.P.H., Ph.D.

Academic Editor

PLOS ONE

Journal Requirements:

2. Please include captions for your Supporting Information files at the end of your manuscript, and update any in-text citations to match accordingly. Please see our Supporting Information guidelines for more information: http://journals.plos.org/plosone/s/supporting-information

Reviewers' comments:

Reviewer's Responses to Questions

**Comments to the Author**

1. Is the manuscript technically sound, and do the data support the conclusions?

Reviewer #1: Partly

Reviewer #2: Partly

2. Has the statistical analysis been performed appropriately and rigorously? 

Reviewer #1: Yes

Reviewer #2: I Don't Know

3. Have the authors made all data underlying the findings in their manuscript fully available?

Reviewer #1: Yes

Reviewer #2: Yes

4. Is the manuscript presented in an intelligible fashion and written in standard English?

Reviewer #1: Yes

Reviewer #2: No

5. Review Comments to the Author

Reviewer #1: My complete review file is uploaded as an attachment.

This manuscript analyzes 650,000 test statistics from 40,000 published psychology articles over almost 40 years to provide an estimation of statistical power, publication bias and p-hacking, and the false discovery rate across the field. To briefly summarize the results, the statistical power was low, but shows a (small) increase over time. There is an excess of statistically significant results, and the false discovery rate across the field is around 32%.

Overall, I think this article does a good job of conveying the issue of an excess of statistically significant results in the psychology field. It is particularly worrisome that so many reported results are statistically significant, and that the statistical power has increased so little over the past few years. The introduction of the article is accessible, gives a clear introduction to topics like publication bias, and provides a comprehensive overview of how these issues relate to problematic consequences within science. I checked out the supplemental materials associated with this manuscript. I did not try to reproduce all results, but it is relatively easy to load in data and to extract specific information from the script. I also appreciate the use of a README file to indicate which files direct me to which results.

This manuscript can benefit from more clarification on certain points to better convey its message. In short, I think the “p-hacking” element of the study deserves more nuance, some robustness checks and other methodological choices could be added, and I believe a redistribution of the information conveyed in the manuscript and supplemental material is warranted. Improving these points can help better facilitate the use of the considerable amount of results, and help readers better assess the methods and results in the study.

I have constructed the remainder of this review into four sections: p-hacking, other methodology, supplemental material, and minor remarks. The numbers at the beginning of each point refer to line numbers in the manuscript. Please feel free to contact me for clarifications.

Signed,

Esther Maassen

My complete review file is uploaded as an attachment.

Reviewer #2: As comments exceed the 20000 characters limit, they are attached as a DOCX file.

Please, find below the introduction of my review:

The author shows a large-scale epidemiological review of the literature in psychology and sociology from APA PsycArticles®, a set of 221,554 articles in 119 peer-reviewed journals. The supporting material is extensive, with R scripts, the study database and a large appendix with supplementary tables and figures. Many robustness analyses and simulations have been performed to assess the impact of hypotheses and chose the best models. The subject is relevant in the reproducibility crisis of science.

However, there are serious limitations inherent to the design of the study; some of them cannot be easily fixed. At least, the limitations should be discussed. Some results may have to be changed or removed. The editor should consider the limitations in his acceptation or refusal of the manuscript.

In addition, there are many minor problems with the actual structure of the manuscript that can be fixed easily: better reporting of Methods, with less reliance to Supplementary Information, a structured Discussion, including a paragraph about strengths and limitations, a comparison to the literature and research perspectives. The editorial entitled “The case for structuring the discussion of scientific papers” (BMJ 1999;318:1224) may help the author.

6. PLOS authors have the option to publish the peer review history of their article (what does this mean?). If published, this will include your full peer review and any attached files.

Reviewer #1: **Yes: **Esther Maassen

Reviewer #2: **Yes: **André GILLIBERT

---

## [Author Response · Author response to Decision Letter 0]

17 Oct 2021

Dear Esther Maassen, dear Dr. André Gillibert, 

Many thanks for your constructive and very helpful comments that I hope improved the paper substantially. Besides the changes reported below, the manuscript was, as suggested, entirely rewritten to assure better readability and at the same time convey more methodological details in the main text. 

In the response.docx file you can find my point by point replies to each of your remarks. The reply text is highlighted in red. Passages that were added in the manuscript and addressed the related issue were highlighted in green. 

I hope that the manuscript is now suitable for publication in PLOS ONE. 

Again many thanks for your remarks and best wishes

Andreas Schneck

---

## [Decision Letter · Decision Letter 1]

11 Nov 2021

PONE-D-21-02988R1Are most published research findings false? Trends in statistical power, publication selection bias, and the false discovery rate in psychology (1975–2017)PLOS ONE

Dear Dr. Schneck,

Thank you for submitting your manuscript to PLOS ONE. After careful consideration, we feel that it has merit but does not fully meet PLOS ONE’s publication criteria as it currently stands. Therefore, we invite you to submit a revised version of the manuscript that addresses the points raised during the review process.

We look forward to receiving your revised manuscript.

Kind regards,

Florian Naudet, M.D., M.P.H., Ph.D.

Academic Editor

PLOS ONE

Reviewers' comments:

Reviewer's Responses to Questions

**Comments to the Author**

1. If the authors have adequately addressed your comments raised in a previous round of review and you feel that this manuscript is now acceptable for publication, you may indicate that here to bypass the “Comments to the Author” section, enter your conflict of interest statement in the “Confidential to Editor” section, and submit your "Accept" recommendation.

Reviewer #2: (No Response)

2. Is the manuscript technically sound, and do the data support the conclusions?

Reviewer #2: Partly

3. Has the statistical analysis been performed appropriately and rigorously? 

Reviewer #2: No

4. Have the authors made all data underlying the findings in their manuscript fully available?

Reviewer #2: Yes

5. Is the manuscript presented in an intelligible fashion and written in standard English?

Reviewer #2: Yes

6. Review Comments to the Author

Reviewer #2: (No Response)

7. PLOS authors have the option to publish the peer review history of their article (what does this mean?). If published, this will include your full peer review and any attached files.

Reviewer #2: **Yes: **André GILLIBERT

---

## [Author Response · Author response to Decision Letter 1]

6 Apr 2022

Dear Prof. Florian Naudet, dear Dr. André Gillibert, 

First, I want to thank you both for your very helpful and inspiring comments that I hope substantially improved the paper. For a detailed response to each remark, see the response to reviewers. 

I hope that the manuscript is now suitable for publication in PLOS ONE, and thank you again for all of your efforts. 

Again, many thanks and best wishes, 

Andreas Schneck

---

## [Decision Letter · Decision Letter 2]

9 May 2022

PONE-D-21-02988R2Are most published research findings false? Trends in statistical power, publication selection bias, and the false discovery rate in psychology (1975–2017)PLOS ONE

Dear Dr. Schneck,

Thank you for submitting your manuscript to PLOS ONE. After careful consideration, we feel that it has merit but does not fully meet PLOS ONE’s publication criteria as it currently stands. Therefore, we invite you to submit a revised version of the manuscript that addresses the points raised during the review process.

First of all I would like to thank André Gillibert for his very useful and important input on this manuscript. I read the comments and really think that those are valuable and helpful. I also measure the workload required by this new round of revisions. In addition, I really think that we are getting closer to a manuscript closer to acceptance. I'm looking forward to reading a new version of the manuscript with detailed answers to the reviewer's comment. Please pay addition to make sure that the main limitations are very clear both in the abstract and in the conclusion of the paper. It is quite rare to have such an in depth review of a manuscript before publication and I applaud this detailed review because I really think that the manuscript is improving at each step. As an author (as part of PLOS One policy : https://journals.plos.org/plosone/s/editorial-and-peer-review-process), you will be asked to make the peer review history public (open peer review). I strongly support this approach -especially for this manuscript- and I definitively hope that you will opt for this. 

We look forward to receiving your revised manuscript.

Kind regards,

Florian Naudet, M.D., M.P.H., Ph.D.

Academic Editor

PLOS ONE

Reviewers' comments:

Reviewer's Responses to Questions

**Comments to the Author**

1. If the authors have adequately addressed your comments raised in a previous round of review and you feel that this manuscript is now acceptable for publication, you may indicate that here to bypass the “Comments to the Author” section, enter your conflict of interest statement in the “Confidential to Editor” section, and submit your "Accept" recommendation.

Reviewer #2: (No Response)

2. Is the manuscript technically sound, and do the data support the conclusions?

Reviewer #2: Partly

3. Has the statistical analysis been performed appropriately and rigorously? 

Reviewer #2: I Don't Know

4. Have the authors made all data underlying the findings in their manuscript fully available?

Reviewer #2: Yes

5. Is the manuscript presented in an intelligible fashion and written in standard English?

Reviewer #2: Yes

6. Review Comments to the Author

Reviewer #2: First, I thank the author to have taken in account all my comments thoroughly and to have deeply reworked the manuscript. I have many new comments (in the attached Word document). I think that the author is closer to a good manuscript.

7. PLOS authors have the option to publish the peer review history of their article (what does this mean?). If published, this will include your full peer review and any attached files.

Reviewer #2: **Yes: **André GILLIBERT

---

## [Author Response · Author response to Decision Letter 2]

16 Oct 2022

Dear Dr. André Gillibert, 

Thank you again for your constructive and helpful comments that substantially improved the paper. Below you find my point-by-point response to your remarks; the newly introduced text is highlighted in blue, and my reply in green. 

I hope that the manuscript is now suitable for publication in PLOS ONE. 

Again many thanks for your remarks and best wishes

Andreas Schneck

---

## [Decision Letter · Decision Letter 3]

12 Dec 2022

PONE-D-21-02988R3Are most published research findings false? Trends in statistical power, publication selection bias, and the false discovery rate in psychology (1975–2017)PLOS ONE

Dear Dr. Schneck,

Thank you for submitting your manuscript to PLOS ONE. After careful consideration, we feel that it has merit but does not fully meet PLOS ONE’s publication criteria as it currently stands. Therefore, we invite you to submit a revised version of the manuscript that addresses the points raised during the review process.

First of all, **I would like to thank the reviewer for his comments**. He kindly ask me for a delay in reviewing the paper because of workload and I agreed so any inconvenience in terms of delays is on my own responsibility. Indeed, his comments seems very important to me. I now think that we are getting closer to acceptance. The few remaining comments don't look very difficult to address but remain quite important. As the reviewer suggests, I consider that because the numerical results may somewhat change, this is a major revision. But please be sure that I really consider this manuscript as an important one and that I am really looking forward to considering your revised paper. 

We look forward to receiving your revised manuscript.

Kind regards,

Florian Naudet, M.D., M.P.H., Ph.D.

Academic Editor

PLOS ONE

Reviewers' comments:

Reviewer's Responses to Questions

**Comments to the Author**

1. If the authors have adequately addressed your comments raised in a previous round of review and you feel that this manuscript is now acceptable for publication, you may indicate that here to bypass the “Comments to the Author” section, enter your conflict of interest statement in the “Confidential to Editor” section, and submit your "Accept" recommendation.

Reviewer #2: (No Response)

2. Is the manuscript technically sound, and do the data support the conclusions?

Reviewer #2: Partly

3. Has the statistical analysis been performed appropriately and rigorously? 

Reviewer #2: Yes

4. Have the authors made all data underlying the findings in their manuscript fully available?

Reviewer #2: Yes

5. Is the manuscript presented in an intelligible fashion and written in standard English?

Reviewer #2: Yes

6. Review Comments to the Author

Reviewer #2: Please, see the attachment for all comments.

Sorry for this late response. I was quite busy lately.

Thank you for the extensive work you performed to answer to my comments.

Almost all problems in the previous manuscript are fixed. The few remaining problems should not be hard to fix. Although my reviewing is lengthy, I think that the issues it rises are not that that big.

There may be a few substantial modifications that may change some results. This is why I consider that there is a “major revision” although it is less major than previous ones.

7. PLOS authors have the option to publish the peer review history of their article (what does this mean?). If published, this will include your full peer review and any attached files.

Reviewer #2: **Yes: **André GILLIBERT

---

## [Author Response · Author response to Decision Letter 3]

12 Jun 2023

Dear Dr. André Gillibert, 

Sorry for the now immense time lag of my revision. Due to my parental leave, I was not able to start the revision before May. 

Thank you once again for your constructive and in-depth comments that improved the paper. You can find my point-by-point response in the response_rev_rev_rev_rev.docx file.

I hope that the manuscript is now suitable for publication in PLOS ONE. 

Again many thanks for your remarks and best wishes

Andreas Schneck

---

## [Decision Letter · Decision Letter 4]

31 Jul 2023

PONE-D-21-02988R4Are most published research findings false? Trends in statistical power, publication selection bias, and the false discovery rate in psychology (1975–2017)PLOS ONE

Dear Dr. Schneck,

Thank you for submitting your manuscript to PLOS ONE. After careful consideration, we feel that it has merit but does not fully meet PLOS ONE’s publication criteria as it currently stands. Therefore, we invite you to submit a revised version of the manuscript that addresses the points raised during the review process.

First of all, I would like to thank again the reviewer for the considerable work on this submission. I'm also impressed by the constructive dialogue between the reviewer and you. I really think that it improved the paper. I'm confident that it helped to improve the manuscript and I am happy to be closer than ever to acceptance. Please address all the minor revisions requested by the reviewer, especially the ones related with possible inconsistencies in the equations. 

We look forward to receiving your revised manuscript.

Kind regards,

Florian Naudet, M.D., M.P.H., Ph.D.

Academic Editor

PLOS ONE

Journal Requirements:

Reviewers' comments:

Reviewer's Responses to Questions

**Comments to the Author**

1. If the authors have adequately addressed your comments raised in a previous round of review and you feel that this manuscript is now acceptable for publication, you may indicate that here to bypass the “Comments to the Author” section, enter your conflict of interest statement in the “Confidential to Editor” section, and submit your "Accept" recommendation.

Reviewer #2: (No Response)

2. Is the manuscript technically sound, and do the data support the conclusions?

Reviewer #2: Yes

3. Has the statistical analysis been performed appropriately and rigorously? 

Reviewer #2: Yes

4. Have the authors made all data underlying the findings in their manuscript fully available?

Reviewer #2: Yes

5. Is the manuscript presented in an intelligible fashion and written in standard English?

Reviewer #2: No

6. Review Comments to the Author

Reviewer #2: The manuscript has been improved and is now quite good in my opinion. The objective, methods, results and discussion are consistent. Limitations are well acknowledged.

I do not have major comments but have many minor comments and suggestions, spotting stylistic issues and minor errors.

Maybe the worst issue lies in probable errors of equations shown in the Appendix. I hope that these equations were not used for computations.

I would like to thank the author for the great patience he had, to resubmit repeatedly his manuscript, responding to all my comments for several rounds.

Please, find all comments in the Attachment!

7. PLOS authors have the option to publish the peer review history of their article (what does this mean?). If published, this will include your full peer review and any attached files.

Reviewer #2: **Yes: **André GILLIBERT

---

## [Author Response · Author response to Decision Letter 4]

26 Sep 2023

Dear Prof. Florian Naudet, dear Dr. André Gillibert,

Again, thank you both for your very helpful and inspiring comments. I hope that I addressed the remaining issues. For a detailed response to each remark, see the response to reviewers.

Besides the smaller edits, I rechecked the code and found a small error in estimating the p-hacking rate. As a result, the p-hacking rate was slightly lower (8.8% instead of 9.9% for the main p-hacking scenario). The text was also edited by a native English-speaking proofreader.

I hope the manuscript is now suitable for publication in PLOS ONE, and thank you for all your efforts.

Again, many thanks and best wishes,

Andreas Schneck

---

## [Editor Report · Decision Letter 5]

27 Sep 2023

Are most published research findings false? Trends in statistical power, publication selection bias, and the false discovery rate in psychology (1975–2017)

PONE-D-21-02988R5

Dear Dr. Schneck,

We’re pleased to inform you that your manuscript has been judged scientifically suitable for publication and will be formally accepted for publication once it meets all outstanding technical requirements.

Kind regards,

Florian Naudet, M.D., M.P.H., Ph.D.

Academic Editor

PLOS ONE
---

## [Editor Report · Acceptance letter]

9 Oct 2023

PONE-D-21-02988R5 

Are most published research findings false? Trends in statistical power, publication selection bias, and the false discovery rate in psychology (1975–2017) 

Dear Dr. Schneck:

I'm pleased to inform you that your manuscript has been deemed suitable for publication in PLOS ONE. Congratulations! Your manuscript is now with our production department. 

Kind regards, 

on behalf of

Pr. Florian Naudet 

Academic Editor

PLOS ONE